# IFI16 Is Indispensable for Promoting HIF-1α-Mediated APOL1 Expression in Human Podocytes under Hypoxic Conditions

**DOI:** 10.3390/ijms25063324

**Published:** 2024-03-15

**Authors:** Richaundra K. Randle, Venkateswara Rao Amara, Waldemar Popik

**Affiliations:** 1Department of Biomedical Sciences, School of Graduate Studies, Meharry Medical College, Nashville, TN 37208, USA; rrandle@mmc.edu; 2Center for AIDS Health Disparities Research, Meharry Medical College, Nashville, TN 37208, USA; venkat.amara@niperhajipur.ac.in; 3Department of Regulatory Toxicology, National Institute of Pharmaceutical Education and Research, Hajipur 844102, Bihar, India; 4Department of Internal Medicine, School of Medicine, Meharry Medical College, Nashville, TN 37208, USA

**Keywords:** podocytes, nephropathy, hypoxia, HIF-1α, hypoxia response element (HRE), APOL1, IFI16, chromatin immunoprecipitation (ChIP)

## Abstract

Genetic variants in the protein-coding regions of APOL1 are associated with an increased risk and progression of chronic kidney disease (CKD) in African Americans. Hypoxia exacerbates CKD progression by stabilizing HIF-1α, which induces APOL1 transcription in kidney podocytes. However, the contribution of additional mediators to regulating APOL1 expression under hypoxia in podocytes is unknown. Here, we report that a transient accumulation of HIF-1α in hypoxia is sufficient to upregulate APOL1 expression in podocytes through a cGAS/STING/IRF3-independent pathway. Notably, IFI16 ablation impedes hypoxia-driven APOL1 expression despite the nuclear accumulation of HIF-1α. Co-immunoprecipitation assays indicate no direct interaction between IFI16 and HIF-1α. Our studies identify hypoxia response elements (HREs) in the APOL1 gene enhancer/promoter region, showing increased HIF-1α binding to HREs located in the APOL1 gene enhancer. Luciferase reporter assays confirm the role of these HREs in transcriptional activation. Chromatin immunoprecipitation (ChIP)–qPCR assays demonstrate that IFI16 is not recruited to HREs, and IFI16 deletion reduces HIF-1α binding to APOL1 HREs. RT-qPCR analysis indicates that IFI16 selectively affects APOL1 expression, with a negligible impact on other hypoxia-responsive genes in podocytes. These findings highlight the unique contribution of IFI16 to hypoxia-driven APOL1 gene expression and suggest alternative IFI16-dependent mechanisms regulating APOL1 gene expression under hypoxic conditions.

## 1. Introduction

Apolipoprotein L1 (APOL1) is a vital component of high-density lipoprotein particles in blood [1,2] and plays a critical role in immunity against African *Trypanosoma brucei* (*T. brucei*) parasites [3,4,5]. Two coding variants of APOL1, G1 and G2, have evolved to confer resistance against the subspecies *T. brucei rhodesiense* [6,7,8]. While these variants offer protection against parasitic infection, their presence in homozygous forms (G1/G1, G1/G2, or G2/G2) is strongly associated with an increased risk of kidney dysfunction and the rapid progression of chronic kidney disease (CKD), particularly in African Americans [9,10,11,12]. These APOL1 risk variants (RVs) are linked to a spectrum of kidney diseases [13,14], including HIV-associated nephropathy [15,16,17], focal segmental glomerulosclerosis [18,19], sickle cell nephropathy [20], and lupus nephritis [21,22,23,24]. The APOL1 variants predominantly affect podocytes, key cells in the kidney glomerulus [25,26,27], leading to compromised glomerular filtration barrier integrity, proteinuria, and eventual end-stage renal disease (ESRD) [28,29].

Despite the prevalence of these high-risk genotypes in approximately 13% of African Americans, not every carrier develops clinically significant renal disease [30]. This discrepancy suggests that APOL1-mediated cytotoxicity [31,32,33,34,35,36,37], believed to manifest only beyond a certain expression threshold, might require additional modifying factors to induce APOL1-associated nephropathy.

A recent study has demonstrated that hypoxia, characterized by low cellular oxygen concentrations, activates APOL1 transcription in kidney podocytes and tubular cells [38]. This finding aligns with the known role of renal hypoxia in kidney injury and repair and the pathogenesis of CKD and ESRD [39,40,41,42,43,44,45,46]. The innate vulnerability of kidneys to hypoxia is attributed to their vascular structure and high-energy-demanding functions [47].

The response to hypoxia at the cellular level involves a shift from oxidative phosphorylation to anaerobic glycolysis, triggering a gene expression cascade regulated by hypoxia-inducible factors (HIFs) to facilitate cell survival in hypoxic conditions [48]. HIF1 is a heterodimeric protein consisting of a constitutively expressed HIF-1β subunit and an oxygen-sensitive HIF-1α subunit [49,50,51,52]. Under normal oxygen tensions, HIF-1α is hydroxylated on proline and asparagine residues. Hydroxylated HIF-1α binds to the von Hippel–Lindau (VHL) protein, targeting the molecule for ubiquitination and degradation by proteasomes [53]. A decreased oxygen tension inhibits prolyl hydroxylase (PHD) and asparaginyl hydroxylase (FIH) activities, resulting in the stabilization of HIF-1α. Stabilized HIF-1α translocates to the nucleus, where it binds in complex with HIF-1β to highly conserved hypoxia response element (HRE) sequences present in the promoter or enhancer regions to drive the transcription of hypoxia-inducible genes [54,55].

Our previous work demonstrated that nucleosomal DNA activates APOL1 expression in human podocytes cultured in normoxic conditions via the cGAS/IFI16/STING/IRF3 signaling pathway [56], a response typically triggered by the detection of pathogenic double-stranded DNA in the cytoplasm [57,58,59]. Interestingly, hypoxia has been shown to induce the release of mitochondrial DNA into the cytoplasm, potentially activating this pathway [60,61] and APOL1 expression.

In this study, we explored the role of the cGAS/IFI16/STING/IRF3 pathway in promoting APOL1 expression in response to hypoxia in human cultured AB8/13 kidney podocytes. Intriguingly, our results revealed that the cGAS/STING/IRF3 pathway does not play a direct role in this process. Instead, our findings highlight the critical yet indirect role of IFI16 in HIF-1α-mediated APOL1 expression. This suggests a more complex regulatory mechanism, potentially implicating IFI16 in epigenetic modifications that modulate chromatin architecture, thereby influencing APOL1 gene expression. These findings may open up new avenues for therapeutic intervention targeting APOL1 expression, particularly in patients carrying APOL1 risk variants.

## 2. Results

### 2.1. A Transient Accumulation of Hypoxia-Induced HIF-1α Is Sufficient to Stimulate the Expression of APOL1 in Human Podocytes

Recent findings suggest that hypoxia stimulates the expression of the APOL1 gene in cultured human podocytes through the interaction of hypoxia-inducible factor (HIF)-1α with a DNA region located approximately 3 kb upstream of the APOL1 promoter [38]. However, the hypoxia response elements (HREs) responsible for APOL1 expression were not identified. To gain further insights into the mechanism driving the hypoxia-induced upregulation of APOL1 expression, we exposed human conditionally immortalized and differentiated AB8/13 podocytes to varying durations of either a hypoxic environment (1% oxygen) or 100 µM of roxadustat, an inhibitor of HIF-prolyl hydroxylase known to stabilize HIF-1α under normoxic conditions. Under both experimental conditions, a noticeable increase in APOL1 mRNA levels was observed (Figure 1A,C). Conversely, the mRNA levels of HIF-1α decreased over time (Figure 1B,D).

To validate the involvement of HIF-1α in the upregulation of APOL1 expression in podocytes, we performed transient transfections of AB8/13 podocytes using a specific pool of siRNA targeting HIF-1α or a non-targeting control siRNA. Following a 48 h transfection, the cells were exposed to roxadustat for six or 24 h (Figure 1E,F). Our findings reveal that a 24 h treatment with roxadustat effectively induced the expression of the APOL1 protein in the podocytes transfected with control siRNA. The increase in APOL1 expression correlated with the accumulation of the HIF-1α protein. Notably, the accumulation of the HIF-1α protein was transient, reaching its peak after a six-hour treatment with roxadustat and subsequently decreasing. In contrast, the knockdown of HIF-1α attenuated the roxadustat-mediated increase in APOL1 protein expression. This underscores the significance of the early and transient stabilization of HIF-1α during hypoxia in driving APOL1 expression in podocytes.

### 2.2. The Minimal Impact of the cGAS/STING/IRF3 Pathway on Roxadustat-Induced APOL1 Expression

Under hypoxic conditions, the potential release of mitochondrial DNA into the cytoplasm may activate the cGAS/IFI16/STING/IRF3 pathway, subsequently promoting APOL1 expression [56]. To explore the involvement of this pathway in the roxadustat-induced upregulation of APOL1 expression, AB8/13 podocytes were transiently transfected with a specific pool of siRNA targeting STING, IRF3, or a non-targeting control siRNA. After a 48 h transfection, the cells were exposed to roxadustat treatment for 24 h. Despite roxadustat inducing an approximately 80% reduction in the expression of the STING and IRF3 proteins and transcripts (Figure 2A–D), the expression of APOL1 remained largely unaffected by the roxadustat treatment.

To assess the impact of cGAS on the modulation of APOL1 expression, AB8/13 podocytes were pre-treated with the specific pharmacological cGAS inhibitor G150 [62] for two hours. The activity of the G150 compound was evaluated using the procedures outlined in the Materials and Methods section. Subsequently, the podocytes were either stimulated with roxadustat (Figure 2E) or exposed to hypoxia (Figure 2F) for 24 h. Despite the prior treatment with G150, there was no observable inhibition in the expression of the APOL1 protein under both conditions. This suggests that cGAS does not significantly contribute to the upregulation of APOL1 expression induced by either roxadustat or hypoxia.

### 2.3. IFI16 Plays a Crucial Role in the Induction of APOL1 Expression in Human Podocytes under Hypoxic Conditions

To gain insights into the mechanism of roxadustat-induced APOL1 expression in AB8/13 podocytes, we investigated the potential role of IFI16, which may operate independently of the cGAS/STING/IRF3 pathway. After a 48 h transfection of the AB8/13 podocytes with specific siRNA targeting IFI16 or a non-targeting siRNA as a control, we subsequently exposed the cells to roxadustat treatment for 24 h and assessed APOL1 expression. Our observations revealed a substantial reduction in APOL1 expression induced by roxadustat upon IFI16 knockdown in the AB8/13 podocytes (Figure 3A,B).

To further elucidate the role of IFI16 in the upregulation of APOL1 expression within hypoxic conditions in AB8/13 podocytes, we used the CRISPR–Cas9 system to generate IFI16 knockout (IFI16KO) podocytes derived from AB8/13 cells. Confirming the successful IFI16 knockout in IFI16KO podocytes (Figure 4A), we exposed parental AB8/13 and IFI16KO podocytes to 1% hypoxia for varying durations of 0, 6, and 24 h. Subsequently, we assessed the expression levels of APOL1 and HIF-1α by Western blotting and RT-qPCR analysis (Figure 4A–C).

The AB8/13 podocytes, when subjected to hypoxic conditions, exhibited a notable increase in APOL1 expression. Conversely, the IFI16KO podocytes demonstrated persistently reduced APOL1 expression, evident in both normoxic (control) conditions and when stimulated by roxadustat, in comparison to the AB8/13 podocytes. Intriguingly, the IFI16KO podocytes revealed slightly elevated levels of HIF-1α expression under normoxic conditions (0 h of hypoxia) compared to the AB8/13 podocytes. However, after a 24 h exposure to hypoxia, both podocyte types exhibited a substantial decline in the transcript and protein levels of HIF-1α.

To validate the functionality of the APOL1 gene in the IFI16KO podocytes, we exposed both the AB8/13 and IFI16KO podocytes to roxadustat, IFNγ, or amidobenzimidazole (diABZI) [63], a potent STING agonist, for 24 h. Subsequently, we assessed APOL1 expression by Western blotting (Figure 4D,E). As expected, roxadustat triggered an increase in APOL1 expression exclusively in the AB8/13 podocytes, confirming the pivotal role of IFI16 in mediating hypoxia-induced APOL1 expression. However, both cell types exhibited a substantial elevation in APOL1 expression in response to IFNγ or diABZI, indicating the functional integrity of the APOL1 gene in IFI16KO podocytes. These findings underscore the crucial involvement of IFI16 in regulating APOL1 expression during hypoxia while also highlighting that alternative signaling pathways retain the ability to activate APOL1 expression even in the absence of IFI16. Overall, our results emphasize the critical role of IFI16 in the upregulation of APOL1 expression in AB8/13 podocytes in hypoxic conditions and suggest that the accumulation of HIF-1α, in the absence of IFI16, is insufficient to upregulate APOL1 expression.

### 2.4. The Interaction between IFI16 and HIF-1α Is Dispensable for the Induction of APOL1 Expression in Hypoxic Podocytes

A recent study demonstrated that HIF-1α recruits IFI16 to the hypoxia response element within the aromatase gene promoter in adipocytes [64]. To investigate whether IFI16 facilitates APOL1 gene expression through an interaction with HIF-1α within the nucleus of hypoxic podocytes, we first examined the cellular localization of HIF-1α and IFI16 in podocytes treated with roxadustat or DMSO (control) for four hours, followed by the preparation of cytosolic and nuclear fractions (Figure 5A).

A Western blot analysis revealed the nuclear localization of IFI16 in both the untreated (control, DMSO) and roxadustat-treated AB8/13 podocytes. As expected, IFI16 was absent in the IFI16KO podocytes. Meanwhile, HIF-1α predominantly accumulated in the nuclear fraction of both the AB8/13 and IFI16KO podocytes in the presence of roxadustat. The specificity of the cytosolic and nuclear fractions was confirmed through α-tubulin and lamin B1 expression, respectively (Figure 5A). To investigate whether IFI16 interacts with HIF-1α in the nuclear lysates of the AB8/13 podocytes under roxadustat or DMSO exposure (normoxia), we conducted co-immunoprecipitation (co-IP) assays using anti-HIF-1α or IgG control antibodies (Figure 5B, left panel). As anticipated, the anti-HIF-1α antibody (but not control IgG) specifically immunoprecipitated endogenous HIF-1α and its nuclear interactor HIF-1β, yet it did not reveal the presence of IFI16 in the anti-HIF-1α precipitates. Moreover, neither anti-HIF-1α nor IgG precipitated the nuclear marker lamin B1. To affirm the absence of an interaction between HIF-1α and IFI16, we performed an inverse co-IP using an antibody against IFI16 (Figure 5B, right panel). This analysis confirmed the presence of IFI16 in the anti-IFI16 precipitates; however, both HIF-1α and HIF-1β were not detectable. In summary, our findings suggest that while IFI16 plays a pivotal role in APOL1 gene expression in hypoxic podocytes, its direct interaction with HIF-1α seems dispensable for this process.

### 2.5. Identification of Active Hypoxia Response Elements within the Regulatory Region of the APOL1 Gene

HIF-1α is a pivotal regulator of gene expression under hypoxic conditions by selectively binding to the consensus 5′-A/GCGTG-3′ sequence, known as hypoxia response elements (HREs), situated within gene promoters or enhancers. A recent identification of an HRE within the enhancer region (coordinates: 36,250,101–36,250,603) of the APOL1 gene using the assay for transposase-accessible chromatin sequencing (ATAC-seq) approach [38] prompted us to explore potential additional HREs in the regulatory elements of the APOL1 gene. Employing the UCSC Genome Browser (hg38), we identified six putative HREs with the consensus 5′-A/GCGTG-3′ motifs in the APOL1 gene, covering the promoter/enhancer region of approximately 3 kb. HREs 1–4 were located on the sense APOL1 gene DNA strand, while HRE5 and HRE6 were on the antisense DNA strand. (Figure 6A).

However, considering that less than 1% of the HRE consensus motifs in gene promoters effectively bind different HIF isoforms [65], not all identified HREs within the APOL1 regulatory DNA may be functionally active. To assess the functionality of these putative HREs, we conducted chromatin immunoprecipitation (ChIP)–qPCR assays using fragmented nuclear DNA from AB8/13 podocytes exposed to either normoxia (control) or roxadustat treatment for 4 h. The results revealed increased HIF-1α occupancy on HRE3, HRE5, and HRE6, with HRE5 displaying the highest occupancy in the roxadustat-treated podocytes compared to the control (Figure 6B).

In exploring the potential recruitment of IFI16 to HREs through an interaction with HIF-1α, a ChIP assay using an anti-IFI16 antibody (Figure 6C) did not show significant enrichment of IFI16 on APOL1 HREs in either the control or roxadustat-treated AB8/13 podocytes. This result and the findings from the immunoprecipitation assays (Figure 5) suggest that the association between IFI16 and HIF-1α is not essential for the upregulation of APOL1 gene expression. To gain further insights into the role of IFI16 in the upregulation of APOL1 gene expression in hypoxic conditions, we examined its potential influence on HIF-1α binding to HREs. The IFI16KO podocytes exposed to roxadustat for 4 h exhibited a notable reduction in HIF-1α binding to HREs, comparable to levels observed under normoxic (control) conditions (Figure 6D). This implies that IFI16 plays a crucial role in facilitating the recruitment of HIF-1α to a specific set of HREs in response to hypoxic stimuli, thereby significantly contributing to the upregulation of APOL1 gene expression in podocytes. In summary, our ChIP-qPCR assays indicate an increased occupancy of HIF-1α on selected HREs following roxadustat treatment. However, IFI16 does not appear to be recruited to these HREs via direct interaction with HIF-1α. This finding suggests the involvement of alternative IFI16-mediated mechanisms in regulating APOL1 gene expression under hypoxic conditions.

To further elucidate the functional significance of HREs in driving APOL1 expression, we cloned APOL1 gene fragments containing HRE5 and HRE6 into the pGL4 luciferase reporter vector. Subsequently, these constructs were co-transfected into HEK293T cells along with the pSV β-galactosidase control vector to normalize for transfection efficiency. After a 24 h incubation, the cells were exposed to either DMSO (control) or roxadustat for eight hours (Figure 6E), and the expression of luciferase and β-galactosidase was analyzed in the cell lysates. We found that the luciferase activity normalized to β-galactosidase in the cells transfected with the HRE5- and HRE6-containing vectors was significantly increased upon exposure to roxadustat, confirming the results obtained from our ChIP-qPCR analysis. Intriguingly, even under normoxic conditions (control), the HRE 5-6 DNA fragment elevated luciferase activity in the HEK293T cells, suggesting that in the absence of HIF-1α in normoxic conditions, other cellular factors may bind to these HREs or their surrounding sequences and modulate APOL1 expression.

### 2.6. IFI16 Does Not Significantly Impact the Expression of a Subset of Hypoxia-Targeted Genes in Podocytes

To assess the broader impact of IFI16 on gene expression in podocytes, we subjected both AB8/13 and IFI16KO podocytes to a 24 h treatment with either roxadustat or DMSO (as a control). Subsequently, we analyzed the mRNA expression levels for four genes commonly upregulated in podocytes in hypoxia and two unaffected by hypoxic conditions (Figure 7).

Our RT-qPCR analysis revealed a significant upregulation of the tested hypoxia-targeted genes, LDHA, LOX, PDK1, and VEGFA, in both the AB8/13 and IFI16KO podocytes treated with roxadustat over 24 h. In contrast, the mRNA expression levels of the non-hypoxia-targeted genes, cGAS and STING, remained essentially unchanged following the roxadustat treatment in both cell lines. These findings strongly suggest a specific role of IFI16 in the potential regulation of APOL1 gene expression in podocytes.

Based on the findings of this study, we propose a model depicting the essential role of nuclear-localized IFI16 in regulating APOL1 gene expression in podocytes during hypoxic conditions (Figure 8).

## 3. Discussion

Arterial blood typically maintains an oxygen tension in the range of 75–100 mmHg, but within a normal kidney, oxygen levels vary significantly, dropping to as low as 5 mmHg in the medulla and 50 mmHg in the cortex [66]. This observation suggests that intermittent fluctuations in blood flow to the kidneys might transiently stabilize HIFs and activate genes regulated by hypoxia in cortical glomerular podocytes. Numerous studies have demonstrated that hypoxia-induced injury of kidney cells, including podocytes, is linked to the accumulation of HIFs [67,68,69] and an accelerated progression of chronic kidney disease (CKD) [70,71]. Significantly, the presence of the APOL1 risk alleles G1 and G2, which are associated with increased susceptibility to podocyte injury, has been implicated in the accelerated progression of chronic kidney disease (CKD) [26]. This observation gains further significance in light of recent discoveries that reveal a marked upregulation of APOL1 expression in renal podocytes under hypoxic stress [38]. These findings underscore the critical interaction between genetic susceptibility and environmental stressors in the pathogenesis of CKD, highlighting the pivotal role of APOL1 in mediating renal vulnerability in hypoxic conditions.

In this study, we have demonstrated that human immortalized and differentiated AB8/13 podocytes, when cultured under hypoxic conditions (1% oxygen) or exposed to the hypoxia mimetic drug roxadustat, display a significant upregulation of APOL1 mRNA and protein levels, consistent with previous observations [38]. Interestingly, we observed a transient accumulation of the HIF-1α protein concurrent with a progressive decline in HIF-1α mRNA levels, implying the presence of a negative feedback mechanism that modulates HIF-1α accumulation in response to hypoxic stress [72,73]. The interplay between the HIF-1α protein and its mRNA levels is a somewhat under-explored aspect of HIF-1α regulatory dynamics. Given that the HIF-1α protein undergoes degradation, continuous mRNA synthesis is vital, especially under sustained hypoxic conditions. In the absence of new mRNA synthesis, HIF-1α protein concentrations would eventually decrease, despite initial stabilization and accumulation. This dynamic suggests a scenario where, although negative feedback might control gene expression during prolonged hypoxia, the early phases of HIF-1α protein accumulation and stability in podocytes might be sufficient to initiate podocyte injury, particularly through the APOL1 pathway. Furthermore, the prolonged stability of HIF-1α mRNA in IFI16KO cells may lead to increased levels of the HIF-1α protein during the initial and intermediate stages of hypoxia (0 h and 6 h). However, in the absence of APOL1 expression, these conditions could be less detrimental to podocytes, which are particularly vulnerable to APOL1 risk variants.

Our results, supported by HIF-1α knockdown experiments, suggest that transient increases in HIF-1α protein levels are sufficient to trigger APOL1 gene expression in podocytes. However, the potential role of other signaling pathways that might collaborate with HIF-1α in regulating APOL1 expression under hypoxic conditions has not yet been explored. Notably, we have previously established that the activation of the cGAS/IFI16/STING/IRF3 signaling axis by cytosolic DNA is a potent inducer of APOL1 gene expression in podocytes under normoxic conditions [56]. To elucidate the influence of hypoxic conditions on this pathway and its consequent effect on APOL1 expression, we used specific siRNAs to achieve the targeted knockdown of STING and IRF3. Despite a marked reduction in both the mRNA and protein levels of STING and IRF3, treatment with roxadustat upregulated the expression of the APOL1 protein to levels comparable to the cells transfected with the control siRNA. Similarly, the inhibition of cGAS using the specific inhibitor G150 [62] did not significantly alter APOL1 expression in the podocytes exposed to either roxadustat or hypoxic conditions. Intriguingly, the siRNA-mediated knockdown of IFI16 notably suppressed APOL1 expression under hypoxic conditions, revealing a critical role of IFI16 in regulating APOL1 gene expression in this context.

To investigate the role of IFI16 in regulating APOL1 expression under hypoxic conditions, we engineered IFI16 knockout (IFI16KO) podocytes from the AB8/13 cell line using the CRISPR–Cas9 system. The subsequent exposure of AB8/13 and IFI16KO cells to hypoxia (1% oxygen) revealed a significant increase in APOL1 expression in the AB8/13 cells. At the same time, the IFI16KO cells exhibited a lower baseline expression of APOL1 and failed to upregulate APOL1 expression in response to the hypoxic stimulus. Intriguingly, the HIF-1α protein showed a similar pattern of accumulation in both the AB8/13 and IFI16KO cells, indicating that in the absence of IFI16, the accumulation of HIF-1α is insufficient to stimulate APOL1 expression under hypoxic conditions. Moreover, both the AB8/13 and IFI16KO podocytes retained the ability to upregulate APOL1 expression in response to IFNγ or the STING agonist diABZI [63] under normoxic conditions. This observation indicates that IFI16 is essential for upregulating APOL1 expression in response to hypoxia but is not required for APOL1 induction by IFNγ or STING.

The essential role of IFI16 in upregulating APOL1 expression raises questions about the mechanisms underlying IFI16 functions in podocytes. The contribution of IFI16 to transcriptional regulation remains incompletely understood, with evidence of both transcriptional activation [74,75,76] and repression [77,78,79,80]. A recent study proposed that IFI16 is recruited to the hypoxia response element (HRE) in the aromatase gene promoter in human adipocytes by interacting with HIF-1α [64]. To investigate the potential interaction between IFI16 and HIF-1α in podocytes, we initially performed subcellular fractionation to establish the localization of IFI16 in AB8/13 podocytes, both under control conditions and following treatment with roxadustat. We found that IFI16 was predominantly present in the nuclear fraction of these cells, irrespective of the treatment condition. Conversely, HIF-1α showed significant nuclear accumulation in the cells exposed to roxadustat, aligning with its expected response to hypoxic stimuli. The subsequent co-immunoprecipitation (co-IP) assays aimed to unravel any potential interaction between IFI16 and HIF-1α in the nuclear fraction of the AB8/13 podocytes. While we successfully co-precipitated HIF-1β with the HIF-1α antibody, verifying the known interaction between these proteins under hypoxic conditions [78,81], IFI16 was not detected in these complexes. This absence was consistent even when employing low-stringency buffers in our co-IP assays, which could have facilitated the detection of weak or transient interactions. Similarly, we did not detect HIF-1α or HIF-1β in complexes precipitated with the IFI16 antibody. Our results suggest that a direct, stable interaction between IFI16 and HIF-1α is not a prerequisite for the induction of APOL1 gene expression in hypoxic podocytes. However, it remains plausible that IFI16 could indirectly modulate the binding of HIF-1α to HREs within the APOL1 promoter, possibly through a mechanism not involving direct protein–protein interactions.

To explore whether IFI16 regulates HIF-1α binding to HREs, we aimed to identify these elements in the APOL1 gene promoter. Although a prior study identified the HIF-1α-binding region within the enhancer sequence of the APOL1 gene [38], it did not specify the exact locations of the HREs. Using the genomic sequence from the UCSC Genome Browser (hg38), we identified six potential A/GCGTG HIF-1α-binding motifs (HREs) within an approximately 3 kb region upstream of the APOL1 transcription start site. Given that less than 1% of the HRE motifs were shown to effectively bind HIF isoforms [65], it was likely that not all the identified HREs would be functionally active. To determine which HREs actively bind HIF-1α, we conducted chromatin immunoprecipitation (ChIP) assays, followed by quantitative PCR (qPCR), on DNA from both the control and roxadustat-treated AB8/13 and IFI16KO podocytes. This analysis revealed a preferential binding of HIF-1α to HRE3 and HRE6, with HRE5 showing the most significant occupancy. In contrast, HRE1, HRE2, and HRE4 appeared less accessible to HIF-1α. Interestingly, HRE1, HRE2, and HRE4 were located on the sense strand upstream of the APOL1 transcription start site, while HRE5 and HRE6 were situated on the antisense strand. These findings agree with a prior study [38], reinforcing the notion that HREs within the enhancer region of the APOL1 gene are critical in modulating its expression during hypoxic conditions.

To validate the co-IP assay findings, which suggested an absence of a direct interaction between IFI16 and HIF-1α, we performed ChIP assays using an IFI16-specific antibody, followed by qPCR analysis on DNA extracted from the control and roxadustat-treated AB8/13 podocytes. Our ChIP-qPCR results revealed no significant enrichment of IFI16 at any of the HREs, indicating that IFI16 does not directly associate with these sites via HIF-1α. However, the lack of IFI16 in the roxadustat-treated IFI16KO podocytes led to a notable reduction in HIF-1α binding to the HREs, reaching levels comparable to those observed under normoxic conditions. This observation suggests an indirect role of IFI16 in modulating the recruitment or binding efficiency of HIF-1α to the HREs in the APOL1 gene promoter. Therefore, while IFI16 may not directly interact with HIF-1α or the HREs, it appears critical in facilitating or enhancing HIF-1α’s transcriptional activity on the APOL1 gene under hypoxic conditions.

To further investigate the functional significance of the HREs located in the enhancer region of the APOL1 gene, DNA fragments encompassing HRE5 and HRE6 were inserted into the pGL4 luciferase reporter vector and transfected into HEK293T cells. We found that the luciferase activity in the cells transfected with the HRE5- and HRE6-containing vectors was significantly increased upon exposure to roxadustat, confirming the results obtained from our ChIP-qPCR analysis. Notably, an unexpected increase in luciferase activity was observed in the control HEK293T cells, suggesting potential interactions of other cellular factors with the HRE5 and HRE6 sequences under normoxic conditions. To investigate this possibility, we propose further experiments involving reporter constructs with individually cloned HRE5 and HRE6 and constructs with progressively truncated adjacent sequences. This approach will enable us to discern the specific contributions of HRE5 and HRE6 and the surrounding genomic context to APOL1 gene expression regulation. While our current findings underscore the critical role of HRE5 in modulating APOL1 gene expression under hypoxic conditions, they also hint at the potential involvement of other regulatory elements or factors that may influence APOL1 expression in a manner not solely dependent on HIF-1α.

Although we have established the critical role of IFI16 in HIF-1α-mediated APOL1 expression in podocytes under hypoxic conditions, the specific mechanism is yet to be determined. To investigate whether IFI16 affects the expression of genes beyond APOL1 in hypoxia, we assessed the expression levels of the LDHA, LOX, PDK1, and VEGFA genes, which are commonly upregulated in podocytes by hypoxia. Our RT-qPCR analysis demonstrated a significant upregulation of these genes in both the AB8/13 and IFI16KO podocytes exposed to roxadustat. However, the expression levels of cGAS and STING remained relatively unchanged in both cell types. These findings suggest that IFI16 may be necessary to regulate the expression of only selected genes, including APOL1. Based on our findings indicating that IFI16 does not directly interact with HIF-1α, we hypothesize that IFI16’s impact on APOL1 expression could be mediated through indirect mechanisms, possibly involving epigenetic modifications. Such modifications could affect chromatin architecture, thereby modulating the accessibility of transcription factors like HIF-1α to their target DNA sequences. IFI16 might alter posttranslational modifications of HIF-1α or its interacting partners [82], potentially enhancing HIF-1α’s affinity for HREs. Considering the known impact of CpG dinucleotide methylation on the binding efficiency of HIF complexes to HREs [83], IFI16 may facilitate the demethylation of these sites, thus promoting HIF-1α binding and subsequent APOL1 gene activation. Additionally, IFI16 could influence histone methylation states by interacting with methyltransferases that enforce transcriptional repression under hypoxic conditions [84,85]. Our future research will focus on dissecting these hypothesized epigenetic mechanisms. This will extend our understanding of IFI16’s regulatory roles in hypoxia-induced gene expression and also aid in identifying potential therapeutic targets to prevent APOL1 expression in podocytes under hypoxic stress.

## 4. Materials and Methods

### 4.1. Cell Culture

Human conditionally immortalized glomerular podocytes AB8/13 [86] were cultured at 33 °C in 1X RPMI 1640 media (Gibco, Carlsbad, CA, USA) containing 10% fetal bovine serum (FBS, GenClone, Genesee Scientific, San Diego, CA, USA), insulin–transferrin–selenium (ITS, Invitrogen, Carlsbad, CA, USA), and 1% penicillin–streptomycin–glutamine (Pen-strep, Invitrogen, 100×). Podocyte differentiation was achieved by transferring the cells from 33 °C to 37 °C for 10–12 days to inactivate temperature-sensitive SV40 large T antigen. HEK293T cells [87,88] were cultured at 37 °C in Dulbecco’s modified Eagle’s medium (DMEM, Gibco) supplemented with 10% FBS and 1% Pen-strep (Invitrogen, 100×).

### 4.2. Generation of IFI16 Knockout (IFI16KO) Cells

Undifferentiated AB8/13 podocytes were cultured in six-well plates (2.5 × 10^5^ cells/well) and transfected with the IFI16 CRISPR double nickase plasmid (1 μg/mL, Santa Cruz Biotechnology, Dallas, TX, USA, sc-416568) using the jetPrime reagent (Polyplus, New York, NY, USA) following the manufacturer’s protocol. After 48 h, the cells were treated with puromycin (5 μg/mL) for 10 days to select permanently transfected cells. The culture media were replaced every 2–3 days with fresh full culture media supplemented with puromycin. Selected IFI16KO podocytes were tested by immunoblotting and RT-qPCR to confirm IFI16 knockout.

### 4.3. Roxadustat Treatment

To induce hypoxia and stabilize HIF-1α in normoxic conditions, HEK293T, AB8/13, and IFI16KO cells were exposed to the HIF-1α prolyl hydroxylase inhibitor roxadustat (Cayman Chemicals, 15294, Ann Arbor, MI, USA) dissolved in DMSO. Then, 100 µM of roxadustat was added to an aliquot of the culture media. The roxadustat-supplemented media were added to cells seeded in six-well plates (3.5–4 × 10^5^ cells/well) or 10 cm dishes (2.4–2.5 × 10^6^ cells/dish) for the indicated time points.

### 4.4. Hypoxia Experiments

To induce physiological hypoxia in AB8/13 and IFI16 KO cells, cells were incubated in a Bactrox Hypoxic/Microaerophilic Chamber (Shel Lab, Cornelius, OR, USA) set at the following conditions: 1% O_2_, 5% CO_2_, and 94% N_2_. Once the chamber reached the desired oxygen concentration, cells were incubated in these conditions for the times indicated. Cellular media and PBS (pH 7.2, Gibco) for cell rinsing were placed inside the chamber during the calibration period to equilibrate the reagents to hypoxic conditions. Cells were harvested for downstream analyses inside the chamber.

### 4.5. siRNA Transfections

AB8/13 podocytes were seeded in six-well plates (2.5 × 10^5^ cells/well) and transfected with pools of siRNAs (Santa Cruz Biotechnologies, Dallas, TX, USA) specifically targeting IFI16 (sc-35633), HIF-1α (sc-35561), STING (sc-92042), and IRF3 (sc-35710) as well as a non-targeting control siRNA (Santa Cruz Biotechnology, sc-36869) using the jetPrime transfection reagent (Polyplus) according to the manufacturer’s instructions. Forty-eight hours after transfection with 10 nM of siRNA, the cells were treated with roxadustat (100 µM) for the indicated time.

### 4.6. Quantitative Real-Time PCR (RT-qPCR)

RNA was extracted from the cells using the Quick RNA Miniprep kit (Zymo Research, Genesee Scientific, San Diego, CA, USA, R1057). RNA preparations were then treated with TURBO DNase (Life Technologies, Carlsbad, CA, USA) and purified according to the manufacturer’s protocol. Following RNA purification, we ensured the integrity of our RNA samples by measuring the A260/280 ratios, which were invariably within the range of 1.8–1.9. cDNA was synthesized using the iScript Advanced cDNA Synthesis Kit (Bio-Rad, Hercules, CA, USA, 1725038). For each 20 µL of reaction mix, either 1.0 or 1.5 µg of RNA was used. Post synthesis, the cDNA was diluted to a final volume of 40 µL using RNase-free water. RT-qPCR was performed on a CFX-96 instrument (Bio-Rad) using 5 µL of cDNA and 15 µL of the SsoAdvanced Universal Inhibitor SYBR Green Super Mix (Bio-Rad, 1725017) per reaction. The qPCR protocol included initial denaturation at 98 °C for 3 min, followed by 40 cycles, each consisting of a denaturation at 98 °C for 15 s and a combined annealing and extension at 60 °C for 30 s. Post amplification, a melting curve analysis was performed over a temperature range of 65–95 °C with increments of 0.5 °C and a dwell time of 5 s at each step. mRNA expression was normalized to β-actin mRNA levels. Additionally, to enhance the reliability of our qPCR results, we included a negative control in each qPCR reaction. Furthermore, we conducted qPCR analysis directly on the purified RNA samples to confirm that any potential residual DNA was not influencing the amplification curves observed with our primers. PCR primers (Life Technologies) used in this study are listed in Table 1 (Section 2.5).

### 4.7. Protein Extraction, Immunoblotting, and Antibodies

Total cell lysates were harvested in RIPA buffer (Santa Cruz Biotechnology, sc-24948A) containing protease and phosphatase inhibitors (Life Technologies). Lysates were then incubated on ice for 35 min and clarified by centrifugation. Protein concentration was measured using the Pierce™ BCA Protein Assay kit (Thermofisher Scientific, Waltham, MA, USA, 23225). Laemmli buffer (Bio-Rad, 1610737/1610747) supplemented with β-mercaptoethanol (Bio-Rad, 1610710) was added to protein lysates. Samples were boiled at 99 °C for 10 min, resolved by 10% SDS-PAGE, and transferred to a nitrocellulose membrane (Bio-Rad, 1620112). The membrane was blocked with 5% milk (Santa Cruz Biotechnology, sc-2324) in 0.1% tris-buffered saline with Tween 20 (TBST-0.1% Tween 20, 20 nM of Tris, 150 nM of NaCl) for one hour and incubated at 4 °C overnight with primary antibodies. The following primary antibodies were used in our experiments to detect proteins of interest: APOL1 (Genentech, South San Francisco, CA, USA, 3.7D6 P80C, 1:2000), α-Tubulin (Santa Cruz Biotechnology, sc-8035), β-actin (Cell Signaling Technology, Danvers, MA, USA, 8457, 1:2000), cGAS (Cell Signaling Technology, 15102, 1:1000), GAPDH (Santa Cruz Biotechnology, sc-47724, 1:5000), HIF-1α (Cell Signaling Technology, 14179, 1:333), HIF-1β (Cell Signaling Technology, 3414, 1:1000) IFI16 (Santa Cruz, sc-8023, 1:300), IRF3 (Cell Signaling Technology, 11904, 1:1000), lamin B1 (Santa Cruz Biotechnology, sc-365214, 1:600), and STING (Cell Signaling Technology, 13647, 1:1000). Nitrocellulose-bound proteins were detected using the Western Bright ECL system (Advansta, San Jose, CA, USA, K-12045-D50). The intensities of the indicated protein bands were quantified by densitometric scanning using ChemiDoc Imaging System with built-in Image Lab Touch Software Security Edition (Bio-Rad) and normalized against the intensities of their respective β-actin protein bands.

### 4.8. Assessing Activity and Treatment with cGAS Inhibitor G150

The cGAS inhibitor G150 (Cayman Chemical, #28318) was prepared as a 10 Mm stock solution in DMSO. We used G150 at a concentration of 10 µM, which was determined to be non-toxic to the cells. To evaluate the activity of G150, we performed an experiment on AB8/13 podocytes. The cells were pre-treated for 2 h with either G150 or DMSO as a control and then transfected with 1 µg of nsDNA for 18 h. This approach was based on our prior research [56], which established that APOL1 expression can be induced in normoxic conditions through the activation of the cGAS/IFI16/STING pathway. The results from our RT-qPCR analysis indicated a notable reduction, approximately 65%, in APOL1 mRNA expression in the cells pre-treated with G150 in contrast to the control group. This significant decrease in the APOL1 mRNA level suggests that G150 effectively inhibited cGAS activity in our study. In the experiments, AB8/13 podocytes were seeded in six-well plates (2.5 × 10^5^ cells/well), and the next day, they were pretreated for 2 h with 10 µM of G150, followed by the addition of 100 µM of roxadustat. Protein lysates were collected after 24 h and analyzed for APOL1, cGAS, and β-actin expression by immunoblotting.

### 4.9. Subcellular Fractionation of AB8/13 and IFI16KO Podocytes and Immunoblotting

Cytoplasmic and nuclear fractions were prepared following the Preparation of Nuclear Extract Protocol in the Nuclear Complex Co-IP Kit (54001, Active Motif, Carlsbad, CA, USA) as per the manufacturer’s instructions with minor modifications. Briefly, differentiated podocytes were cultured in 10 cm dishes at 37 °C in full RPMI 1640 media (Gibco) containing 10% FBS and ITS. DMSO (control) and roxadustat-treated cells were harvested in ice-cold PBS (pH 7.2, Gibco) containing phosphatase inhibitors. Following centrifugation, the supernatant was discarded, and the cell pellet was resuspended in an ice-cold hypotonic buffer supplemented with 20 mM of sodium butyrate. Following incubation on ice, detergent was added, and the suspension was centrifuged to yield cytoplasmic (supernatant) and nuclear (pellet) fractions. Nuclear pellets were resuspended in the provided digestion buffer, supplemented with 100 mM of PMSF and a protease inhibitor cocktail. An enzymatic shearing cocktail was added to the suspension, followed by EDTA to stop the reaction. The suspension was centrifuged, and the resulting supernatant was saved as the nuclear extract. Protein concentration was quantified in both fractions using the Pierce™ BCA Protein Assay kit (Thermofisher Scientific, 23225). Cytosolic and nuclear extracts were resolved by 10% SDS-PAGE, transferred to a nitrocellulose membrane (Bio-Rad, 1620112), and analyzed by immunoblotting as previously described.

### 4.10. Nuclear Co-Immunoprecipitation of HIF-1α and IFI16 and Immunoblotting

Co-immunoprecipitation assays were performed following the Co-Immunoprecipitation Protocol in the Nuclear Complex Co-IP Kit (54001, Active Motif, Carlsbad, CA, USA) as per manufacturer’s instructions. Briefly, nuclear lysates were prepared from AB8/13 podocytes untreated or treated with roxadustat (100 µM) as described above. A fraction of the nuclear lysate was saved to serve as input. The remaining nuclear lysates were incubated with either a control normal rabbit anti-IgG (Santa Cruz Biotechnology, sc-2027), anti-HIF-1α (Cell Signaling Technologies, 36919), or anti-IFI16 (Santa Cruz Biotechnology, sc-8023) overnight at 4 °C. The nuclear protein complexes were precipitated using protein A/G plus agarose beads (Santa Cruz Biotechnology, sc-2003). Input samples and beads were boiled in the provided 2X Reducing Loading Buffer (130 mM Tris pH 6.8, 4% SDS, 0.02% Bromophenol blue, 100 mM of DTT, 20% glycerol) at 99 °C for 5 min. Samples were resolved by 10% SDS-PAGE, transferred to a nitrocellulose membrane (Bio-Rad, 1620112), and analyzed by immunoblotting as previously described.

### 4.11. Chromatin Immunoprecipitation (ChIP)-qPCR

Podocytes were cultured in 10 cm dishes at 33 °C in full RPMI 1640 media supplemented with 10% FBS (Clontech), ITS (Invitrogen), and 1% Pen-strep for 18–24 h, achieving approximately 80% confluency. Subsequently, podocytes were treated with either DMSO (control) or roxadustat (100 µM) for 4 h. Following the treatment, chromatin immunoprecipitation (ChIP) was conducted using the Pierce Magnetic ChIP Kit (Thermofisher Scientific, 26157) according to the manufacturer’s instructions. Briefly, cellular protein–DNA complexes were crosslinked using 1% formaldehyde. The chromatin was then digested with MNase and released from nuclei by sonication (Branson Sonifier 450, Marshall Scientific, Hampton, NH, USA) on ice, with specific settings of 1W output control and 30% duty cycle for ten 15-s cycles. To prevent chromatin degradation, cycles were spaced out by 1 min. A 1% fraction of the final volume of digested chromatin was preserved for use as the input control. The immunoprecipitation of protein–DNA complexes was carried out using anti-HIF-1α (Cell Signaling Technologies, 36169) or anti-IFI16 (Santa Cruz Biotechnology, sc-8023), and the complexes were isolated utilizing Pierce Protein A/G Magnetic Beads. Antibodies targeting RNA Polymerase II and normal rabbit IgG (provided by Pierce) served as positive and negative controls, respectively. After the reversal of crosslinking and protein digestion by Proteinase K, the resulting DNA fragments were purified and quantified by qPCR. The cycling conditions involved denaturation at 98 °C for 3 min, followed by 40 cycles of denaturation at 98 °C for 15 s, annealing, and extension at 60 °C for 30 s. The synthesis of qPCR amplicons of the expected size was confirmed through melting curve analysis. The primer sequences used in the experiments are detailed in Table 1.

### 4.12. Cloning of APOL1 Hypoxia Response Elements into pGL4 Luciferase Reporter Vector and Luciferase Reporter and β-Galactosidase Assays

APOL1 sequences were obtained from the UCSC Genome Browser. APOL1 HRE sequences were amplified by PCR with Platinum™ II Taq Hot-Start DNA Polymerase (Invitrogen, 14966001) using genomic DNA (gDNA) isolated from AB8/13 podocytes as a template. gDNA was extracted from the cells using the Quick-DNA Miniprep Plus Kit (Zymo Research, D4068). Amplicons were separated by size by agarose gel electrophoresis, and the amplicon of interest was excised from the gel. DNA was extracted using the Zymoclean Gel DNA Recovery Kit (Zymo Research, D4007). A pGL4.10[*luc2*] vector (Promega, Madison, WI, USA, E6651) and the DNA fragments of interest were digested with KpnI-HF (New England Biolabs, Ipswich, MA, USA, R3142S) and NheI-HF (New England Biolabs, R3131S) restriction enzymes, then purified using the DNA Clean and Concentrator-5 Kit (Zymo Research, D4004). Ligation of the insert (150 ng) into the vector (50 ng) was carried out using T4 DNA Ligase (New England Biolabs, M0202T) to yield the pGL4-APOL1 HRE constructs. The primer sequences for the amplification and cloning of APOL1 HREs can be found in Table 1. Constructs were transformed into NEB 10-beta-competent E. coli cells (New England Biolabs, C3019I) following the High-Efficiency Transformation Protocol per the manufacturer’s instructions. Transformed bacteria were grown on a selection plate overnight at 37 °C. Plasmid DNA was extracted from bacterial clones using the Zyppy Plasmid Miniprep Kit (Zymo Research D4020), then digested with KpnI-HF and NheI-HF restriction enzymes and electrophoresed on an agarose gel to confirm incorporation of the APOL1 HRE inserts. APOL1 promoter activity was measured in HEK293T cells using the Steady-Glo Luciferase Assay Kit (Promega, E2510) according to the manufacturer’s instructions with minor modifications. Briefly, cells were cultured in 6-well plates at a density of 3.5–4 × 10^5^ cells per well at 37 °C in DMEM (Gibco) supplemented with 10% FBS and 1% Pen-strep overnight. The next day, cells were transfected with 1 µg of APOL1 HRE reporter vectors, 1 µg of pSV β-galactosidase vector (E1081, Promega), or an empty pGL4.10 [*luc2*] vector using polyethyleneimine (PEI, Polysciences, Warrington, PA, USA). After 16 h, cells were treated with 100 µM of roxadustat as described above. After 8 h, the cellular media were aspirated, rinsed with PBS (pH 7.2, Gibco), and harvested in 1X Glo Lysis Buffer (E266A, Promega). Cell lysates were centrifuged at 13,500 rpm for 15 min to clarify the sample. Lysates were transferred to a black 96-well plate (3915, Costar, ThermoFischer Scientific, Waltham, MA, USA), and an equal volume of the Steady-Glo Assay Reagent (E2510, Promega) was added to each well. After 5 min of incubation at room temperature, luminescence was measured with a luminometer (BioTek Synergy HT, Marshall Scientific, Hampton, NH, USA) with the following settings: sensitivity—200; integration time—3 s; and probe offset—1mm. For the β-galactosidase assay, the Beta-Glo Assay System (E4720, Promega) was used per the manufacturer’s instructions. Samples were incubated at room temperature for thirty minutes, and luminescence was measured with a luminometer (BioTek Synergy HT) with the following settings: sensitivity—135; integration time—0.3–1 s; and probe offset—1 mm. Luciferase activity was normalized to β-galactosidase expression.

### 4.13. Statistical Analysis

In our statistical analysis, we used GraphPad Prism 10, which inherently incorporates tests for homogeneity of variances as part of the ANOVA procedure, using either the Brown–Forsythe test or Bartlett’s test. Additionally, GraphPad Prism provides four distinct tests for assessing normality. Prior to conducting a regular one-way ANOVA, we rigorously verified that our data conformed to both normality and homogeneity of equal variance criteria. Experimental values are expressed as means ± SD of three biological replicates. A comparison of the two groups was performed using an unpaired *t*-test. A comparison of multiple groups was carried out by a one-way ANOVA followed by a post hoc Tukey test.

## Figures and Tables

**Figure 1 ijms-25-03324-f001:**
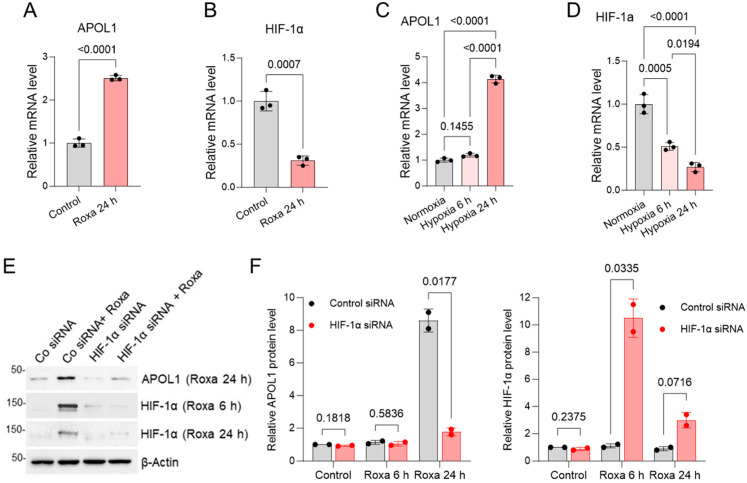
Hypoxia stimulates APOL1 gene expression in human AB8/13 podocytes through the transient accumulation of HIF-1α. (**A**,**B**) AB8/13 podocytes were treated with either DMSO (control) or 100 µM of roxadustat (Roxa) for 24 h. Quantitative real-time PCR (RT-qPCR) analysis was performed to assess the expression levels of APOL1 and HIF-1α mRNA. Data are expressed as means ± SD from three biological replicates. (**C**,**D**) RT-qPCR analysis of APOL1 and HIF-1α mRNA in AB8/13 podocytes subjected to hypoxia (1% oxygen) for the indicated durations. Control cells were maintained under normoxic conditions and collected after 24 h. mRNA expression levels were normalized to β-actin mRNA expression. Values are expressed as means ± SD of three independent biological replicates using one-way ANOVA with a post hoc Tukey test. (**E**) AB8/13 podocytes were transfected with control siRNA (Co) or a siRNA pool targeting HIF-1α mRNA for 48 h, followed by treatment with 100 µM of roxadustat for 6 or 24 h. Immunoblotting was performed to analyze the expression of the HIF-1α and APOL1 proteins, with β-actin protein levels as the loading control. A representative immunoblot image from two independent experiments is shown. Protein ladder markers (kDa) are indicated. (**F**) Quantification of APOL1 (left panel) and HIF-1α (right panel) protein content in AB8/13 podocytes transfected with control siRNA or HIF-1α siRNA followed by roxadustat treatment, as described in (**E**). Intensities of protein bands were quantified by densitometric scanning. Expression levels of APOL1 and HIF-1α proteins were normalized against β-actin levels. The relative APOL1/β-actin and HIF-1α/β-actin ratios were set as 1.0 in control cells exposed to DMSO only. Values are expressed as means ± SD of two independent biological replicates using an unpaired *t*-test.

**Figure 2 ijms-25-03324-f002:**
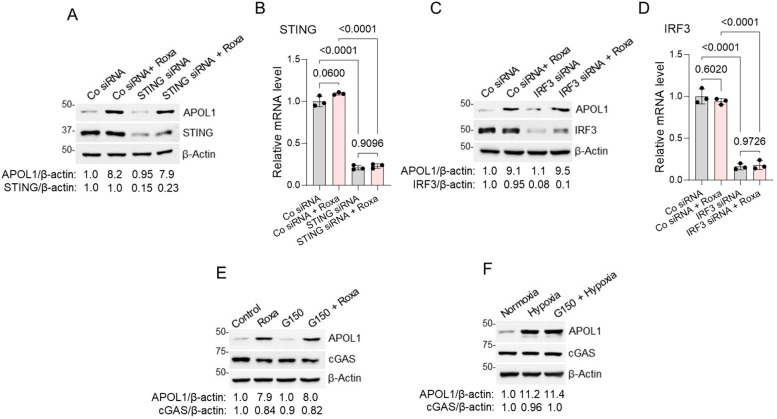
Knockdown of STING, IRF3, or pharmacological inhibition of cGAS does not prevent APOL1 expression in hypoxic conditions. (**A**) AB8/13 podocytes were transfected for 48 h with control siRNA (Co) or siRNA pool targeting STING and subsequently treated with 100 µM of roxadustat (Roxa) for 24 h. Expressions of STING and APOL1 proteins were analyzed by immunoblotting. β-actin protein levels served as the loading control. A representative immunoblot image from two independent experiments is shown. Protein ladder markers (kDa) are indicated. Intensities of protein bands were quantified by densitometric scanning. Expression levels of APOL1 and STING proteins were normalized against β-actin levels. In cells transfected with control siRNA only, the APOL1/β-actin and STING/β-actin ratios were both set as 1.0. (**B**) Expression of STING mRNA was analyzed by RT-qPCR and normalized to β-actin mRNA levels. Values are expressed as means ± SD of three independent biological replicates using one-way ANOVA with a post hoc Tukey test. (**C**) AB8/13 podocytes were transfected for 48 h with control siRNA (Co) or siRNA pool targeting IRF3 and subsequently treated with 100 µM of roxadustat for 24 h. The expression of the IRF3 and APOL1 proteins was analyzed by immunoblotting. β-actin protein levels served as the loading control. A representative immunoblot image from two independent experiments is shown. Protein ladder markers (kDa) are indicated. Intensities of protein bands were quantified by densitometric scanning. Expression levels of APOL1 and IRF3 proteins were normalized against β-actin levels. In cells transfected with control siRNA only, the APOL1/β-actin and IRF3/β-actin ratios were both set as 1.0. (**D**) Expression of IRF3 mRNA was analyzed by RT-qPCR and normalized to β-actin mRNA levels. Values are expressed as means ± SD of three independent biological replicates using one-way ANOVA with a post hoc Tukey test. (**E**) AB8/13 podocytes were exposed to a 2 h pretreatment with 10 µM of the cGAS inhibitor G150, followed by the addition of 100 µM of roxadustat for 24 h. Expression of APOL1 and cGAS protein levels was analyzed by immunoblotting. β-actin protein levels served as the loading control. A representative immunoblot image from two independent experiments is shown. Protein ladder markers (kDa) are indicated. Expression levels of the APOL1 and cGAS proteins (densitometry) were normalized against β-actin levels. In control cells (DMSO), the APOL1/β-actin and cGAS/β-actin ratios were both set as 1.0. (**F**) AB8/13 podocytes were cultured under normoxia or hypoxia (1% oxygen) without G150 or after a 2 h pretreatment with a cGAS inhibitor. Expression of APOL1, cGAS, and HIF-1α protein levels was analyzed by immunoblotting. β-actin protein levels served as the loading control. A representative immunoblot image from two independent experiments is shown. Protein ladder markers (kDa) are indicated. Expression levels of APOL1 and cGAS proteins (densitometry) were normalized against β-actin levels. In control cells (normoxia), the APOL1/β-actin and cGAS/β-actin ratios were both set as 1.0.

**Figure 3 ijms-25-03324-f003:**
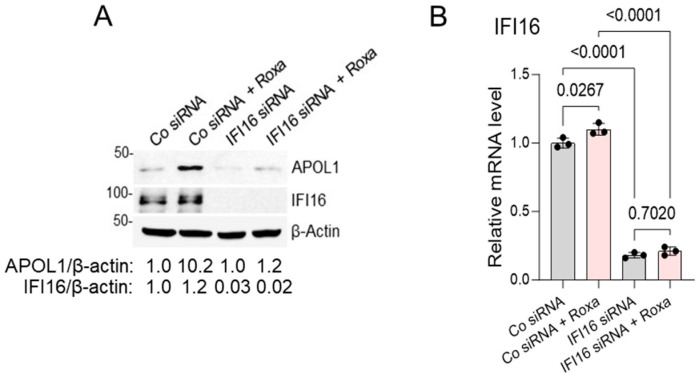
IFI16 knockdown reduces APOL1 gene expression induced by roxadustat in AB8/13 podocytes. (**A**) AB8/13 podocytes were transfected for 48 h with control siRNA (Co) or siRNA pool targeting IFI16 and subsequently treated with 100 µM of roxadustat (Roxa) for 24 h. Expression of IFI16 and APOL1 proteins was analyzed by immunoblotting. β-actin protein levels served as the loading control. A representative immunoblot image from two independent experiments is shown. Protein ladder markers (kDa) are indicated. Intensities of protein bands were quantified by densitometric scanning. Expression levels of APOL1 and IFI16 proteins were normalized against β-actin levels. In cells transfected with control siRNA only, the APOL1/β-actin and IFI16/β-actin ratios were both set as 1.0. (**B**) Expression of IFI16 mRNA was analyzed by RT-qPCR and normalized to β-actin mRNA levels. Values are expressed as means ± SD of three independent biological replicates using one-way ANOVA with a post hoc Tukey test.

**Figure 4 ijms-25-03324-f004:**
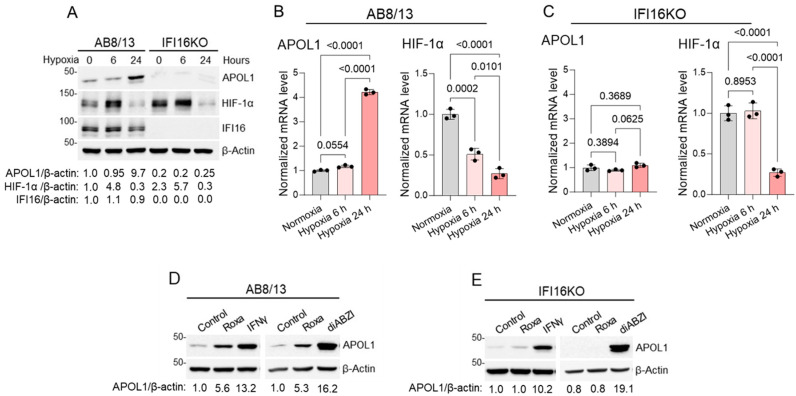
IFI16 knockout attenuates APOL1 gene expression in podocytes. (**A**) AB8/13 and IFI16 knockout (IFI16KO) cells generated from parental AB8/13 podocytes were exposed to hypoxia (1% oxygen) for 0, 6, or 24 h. Expression of APOL1, HIF-1α, and IFI16 proteins was analyzed by immunoblotting. β-actin protein levels served as the loading control. A representative immunoblot image from two independent experiments is shown. Protein ladder markers (kDa) are indicated. Intensities of protein bands were quantified by densitometric scanning. Expression levels of APOL1, HIF-1α, and IFI16 proteins were normalized against β-actin levels. In cells exposed to hypoxia for 0 h, the APOL1/β-actin, HIF-1α/β-actin, and IFI16/β-actin ratios were set as 1.0. (**B**) Expression of APOL1 and HIF-1α mRNA in AB8/13 and (**C**) IFI16KO podocytes exposed to hypoxia (1% oxygen) for the times indicated was analyzed by RT-qPCR and normalized to β-actin mRNA levels. Control cells were incubated in normoxic conditions and collected after 24 h. Values are expressed as means ± SD of three independent biological replicates using one-way ANOVA with a post hoc Tukey test. (**D**) AB8/13 or (**E**) IFI16KO podocytes were treated for 24 h with either DMSO (control), 100 µM of roxadustat, 1 µM of the STING agonist diABZI, or 10 ng/mL of IFNγ. Expression of APOL1 protein levels was analyzed by immunoblotting. β-actin protein levels served as the loading control. A representative immunoblot image from two independent experiments is shown. Protein ladder markers (kDa) are indicated. In the control AB8/13 (**D**) and IFI16KO (**E**, left panel) cells exposed to DMSO only, the APOL1/β-actin ratios were set as 1.0.

**Figure 5 ijms-25-03324-f005:**
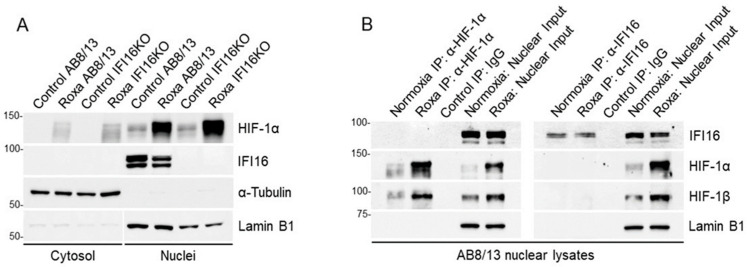
HIF-1α does not interact with IFI16 in the nucleus of roxadustat-treated podocytes. (**A**) AB8/13 and IFI16KO podocytes were treated with 100 µM of roxadustat (Roxa) for 4 h, and then cellular lysates were processed to yield cytosolic and nuclear fractions. Expression of HIF-1α, IFI16, a-tubulin, and lamin B1 proteins was measured by immunoblotting. α-tubulin and lamin B1 proteins indicate cytosolic and nuclear fractions, respectively. A representative immunoblot image from two independent experiments is shown. Protein ladder markers (kDa) are indicated. (**B**) Nuclear fractions isolated from AB8/13 podocytes treated with 100 µM of roxadustat for 4 h were co-immunoprecipitated using anti-IFI16, anti-HIF-1α, or IgG control antibodies. Input (20 µg) and immunoprecipitated (500 µg) samples were analyzed by immunoblotting. A representative immunoblot image from two independent experiments is shown. Protein ladder markers (kDa) are indicated.

**Figure 6 ijms-25-03324-f006:**
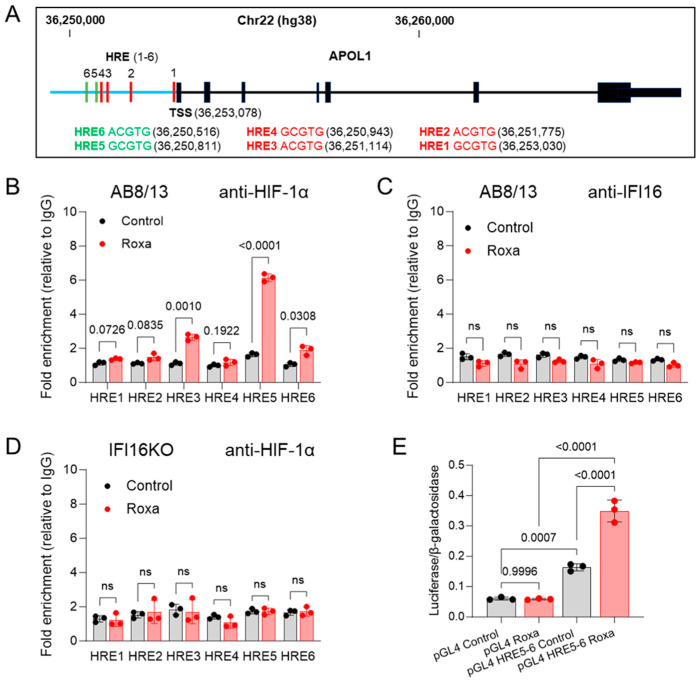
HIF-1α interacts with selected HRE sequences localized in APOL1 regulatory elements. (**A**) Diagram of APOL1 gene locus (UCSC Genome Browser hg38 chr22). HREs 1-4 (red bars) are localized on a plus DNA strand. HREs 5 and 6 (green bars) are localized on a minus strand. APOL1 exons are shown as black bars. The blue horizontal line represents the APOL1 promoter/enhancer region upstream of the transcription start site, TSS. The nucleotide numbering of putative HREs is based on a positive strand of the APOL1 gene. (**B**) AB8/13 podocytes were treated with 100 µM of roxadustat for 4 h and then subjected to chromatin immunoprecipitation using anti-HIF-1α or (**C**) anti-IFI16 antibodies. (**D**) IFI16KO podocytes were treated with 100 µM of roxadustat for 4 h and then subjected to chromatin immunoprecipitation using anti-HIF-1α antibodies. The immunoprecipitated DNA fragments were analyzed by qPCR using primers specific for HRE sequences (Table 1). (**E**) HEK293T cells were co-transfected with an empty pGL4 luciferase vector (pGL4) or with a cloned HRE 5-6 region (pGL4 HRE 5-6) and a control pSV β-galactosidase vector. At 24 h post-transfection, the cells were exposed to 100 µM of roxadustat or DMSO (control) for 8 h and lysed, and the luciferase/β-galactosidase ratio was determined. Values in (**B**–**D**) are expressed as means ± SD of three independent biological replicates using an unpaired *t*-test; ns, not significant (*p* > 0.05). Values in (**E**) are expressed as means ± SD of three independent biological replicates using one-way ANOVA with a post hoc Tukey test.

**Figure 7 ijms-25-03324-f007:**
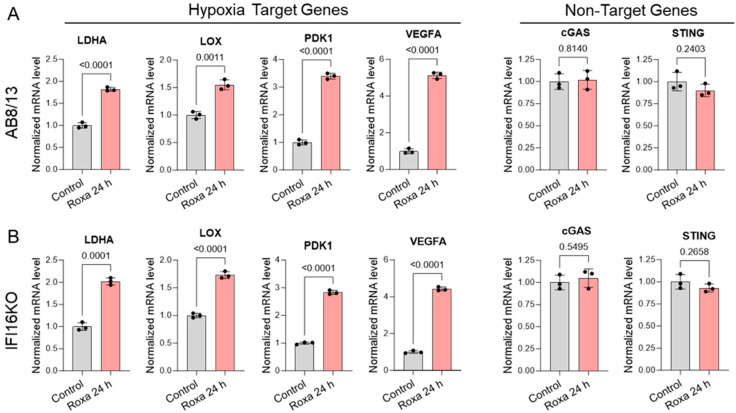
IFI16 knockout does not affect the expression of selected hypoxia-targeted genes in podocytes. (**A**) AB8/13 and (**B**) IFI16KO podocytes were exposed to DMSO (Control) or 100 µM of roxadustat (Roxa) for 24 h. mRNA expression levels of lactate dehydrogenase A (LDHA), lysyl oxidase (LOX), pyruvate dehydrogenase kinase 1 (PDK1), vascular endothelial growth factor A (VEGFA), cGAS, and STING were analyzed by RT-qPCR and normalized to β-actin mRNA levels. Values are expressed as means ± SD of three independent biological replicates using an unpaired *t*-test.

**Figure 8 ijms-25-03324-f008:**
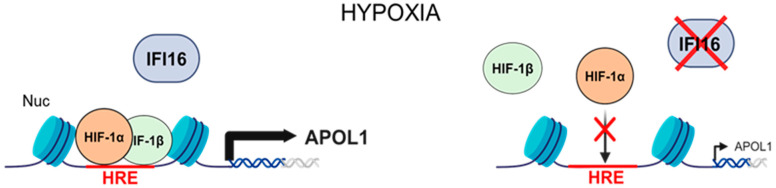
Proposed model illustrating the critical role of the nuclear IFI16 protein in promoting the HIF-1α-mediated transcriptional regulation of the APOL1 gene in podocytes under hypoxia. Under low-oxygen conditions (hypoxia), nuclear IFI16 assists in the binding of the HIF-1α/HIF-1β heterodimer to the hypoxia response element (HRE) within the APOL1 gene’s promoter/enhancer region, thereby facilitating the stimulation of APOL1 gene expression. Conversely, in the absence of IFI16, the capacity of HIF-1α to bind to the HRE and enhance APOL1 gene expression is markedly reduced, underscoring IFI16’s essential role in the HIF-1α-mediated transcriptional activation of the APOL1 gene. The hypothetical positioning of nucleosomes (Nuc) on the APOL1 promoter/enhancer is depicted. This illustration was created with BioRender.com.

**Table 1 ijms-25-03324-t001:** Primer sequences (5’-3’) used in this study.

Primer sequences used for RT-qPCR			
**Gene**	**Forward**	**Reverse**	
VEGFA	TTGCCTTGCTGCTCTACCTCCA	GATGGCAGTAGCTGCGCTGATA	
LDHA	GGATCTCCAACATGCAGCCTT	AGACGGCTTTCTCCCTCTTGCT	
PDK1	CATGTCACGCTGGTAATGAGG	CTCAACACGAGGTCTTGGTGCA	
LOX	ATTTCTTACCCAGCCGACCA	ACTTGCTTTGTGGCCTTCAG	
cGAS	AGGAAGCAACTACGACTAAAGCC	CGATGTGAGAGAAGGATAGCCG	
STING	CCTGAGTCTCAGAACAACTGCC	GGTCTTCAAGCTGCCCACAGTA	
IRF3	TCTGCCCTCAACCGCAAAGAAG	TACTGCCTCCACCATTGGTGTC	
APOL1	GCTTTGCTGAGAGTCTCTGTCC	GGGCTTACTTTGAGGATCTCCAG	
HIF-1α	TATGAGCCAGAAGAACTTTTAGGC	CACCTCTTTTGGCAAGCATCCTG	
IFI16	GATGCCTCCATCAACACCAAGC	CTGTTGCGTTCAGCACCATCAC	
β-ACTIN	CACCATTGGCAATGAGCGGTTC	AGGTCTTTGCGGATGTCCACGT	
Primer sequences used for ChIP-qPCR			
**APOL1HREs**	**Forward**	**Reverse**	**Amplicon Size**
HRE1	GAGGGTGGGGCTGGACTGAA	ACCGAGGAATTCGAAAGGGAAAGTG	103 bp
HRE2	TCCTGGGTGCATCCTCAACCT	CAGCAACTCAGGGAGGAGGC	145 bp
HRE3	TGAGCCGAGATCACAACAC	GGGTCCCAAGATTTGATTTTCC	127 bp
HRE4	CACAACGCCAAGAGATCGAG	CCTCAGCCTCCCAAGTAGC	124 bp
HRE5	TCAGACCAAACTCGTGACCA	TCTCGATCTCTTGGCGTTGT	150 bp
HRE6	GAAACTCCCTGCCCTGTTCT	TCCCTGAGTGCACAATTTCTG	153 bp
Primer sequences used for cloning APOL1 HRE5 and HRE6 into pGL4-luciferase vector			
**Forward (KpnI underlined)**		**Reverse (NheI underlined)**	**Amplicon Size**
gatcGGTACCCCTGACGACCTGTGTATGCA		actgGCTAGCCTCGATCTCTTGGCGTTGTG	387 bp

## Data Availability

The datasets generated from this study are available from the corresponding author upon reasonable request.

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
