# Peer review of "IFI16 Is Indispensable for Promoting HIF-1α-Mediated APOL1 Expression in Human Podocytes under Hypoxic Conditions"

_ijms, 2024, doi:10.3390/ijms25063324_

Round 1

Reviewer 1 Report

Comments and Suggestions for Authors

In the present manuscript submitted by Randle R. K and colleagues, investigated the role of the cGAS/IFI16-STING-IRF3 pathway in promoting APOL1 expression in response to hypoxia. This was done using in vitro culture using conditionally immortalized human glomerular podocytes and HEK293T cells used for APOL1 promoter activity. Based on the resulted presented, authors found unknown mechanism that IFI16 play a role in HIF-1α-mediated APOL1 expression by regulating the epigenetic modifications. The current manuscript written very well and experimental design very appropriate and suitable for proposed experiments.  Authors have previously published (Davis S.E et al Scientific Reports 2019) and revealed the role of IFI16 in HIF-1α-mediated APOL1 expression in podocytes under hypoxic conditions. In current manuscript, authors extended by investigating the IFI16 role in epigenetic modifications to mediate the APOL1 expression under hypoxic condition. The presented results conclude the authors hypothesis and current version can be acted for publication. However, I have minor comments as it is interesting to mimic the findings in using in vivo model of kidney disease would be greatly impact.  

Reviewer 2 Report

Comments and Suggestions for Authors

The manuscript by Randle et al. reports a careful study of the role of IFI16 as a co-activator of HIF-1a-induced apolipoprotein L1 gene expression and protein content in the AB8/13 differentiated human podocyte cell line.  The investigators examined the impact of IFI16 knockout on APOL1 induction by hypoxia (1% O2) vs. roxadustat, an inhibitor of HIF-1a proline hydroxylation.   The experiments demonstrated IFI16 to be essential for hypoxia- and roxadustat-induction of APOL1, but not for APOL1 induction by the STING-IRF3 pathway, showing the latter pathway to activate APOL1 expression independently of HIF-1 induction.  Interestingly, IFI16 knockout did not attenuate hypoxia- and roxadustat-induced HIF-1a nuclear transmigration and DNA binding, nor did IFI16 co-immunoprecipitate with HIF-1a and HIF-1b, suggesting the essential contribution of IFI16 to APOL1 expression is not as a coactivator at the hypoxia-response elements, but in a more indirect fashion.

This study addresses a fundamental mechanism of hypoxia-induced expression of APOL1, a component of high-density lipoproteins implicated in the pathogenesis of chronic kidney disease.  The study systematically examined the interactions of HIF-1a and IFI16 in a differentiated human podocyte cell line.  Important insights on these interactions emerged.  The manuscript is well-written and coherently presents complex molecular signaling mechanisms.  The major comments identify some unresolved questions; forcefully addressing these concerns will increase further the manuscript’s impact.

MAJOR COMMENTS

1.     Information is missing from Figure 1 which would support the proposed relationship between APO1 and HIF-1a contents: [A] Although HIF-1a mRNA abundance and protein content are shown at both 6 and 24 h roxadustat exposure, APOL1 mRNA and protein are shown only at 24 h roxadustat treatment.  The immunoblot (Figure 1E) shows robust HIF-1a content at 6 h roxadustat exposure; was APOL1 content similarly increased at the 6 h mark?  Or is the increase in APOL1 delayed, similar to APOL1 mRNA during hypoxia exposure (Figure 1C)?  [B] Did APOL1 and HIF-1a mRNA abundances follow similar time courses during roxadustat exposure as in hypoxia (Figure 1C, D)?  Did APOL1 and HIF-1a protein contents respond to hypoxia in a manner similar to their responses to roxadustat (Figure 1E)?  [C] Only single immunoblots are provided to show changes in APOL1 and HIF-1a protein contents.  Was the experiment done three times, as is the case for other variables?  Please add a bar graph reporting APOL1 and HIF-1a protein contents at 6 and 24 h control and roxadustat exposure in podocytes transfected with non-targeting control vs. HIF-1a targeting siRNA.

2.     In Figures 2-5, were the immunoblots repeated three times, as were the mRNA measurements?  To strengthen your case, please include bar graphs to accompany the immunoblots.

3.     The cGAS inhibitor G150 had no effect (Figure 2E, F).  While this outcome is concordant with the argument that the cGAS-STING signaling pathway is not required for HIF-1a mediated APOL1 formation, it is important to demonstrate the applied G150 concentration did indeed inhibit cGAS.

4.     IFI16 knockout delayed the hypoxic decline of HIF-1a mRNA abundance (Figure 4 B vs. C), even though HIF-1a protein content appeared to be unaffected by IFI16 knockout (Figure 4A).  Do these findings have implications for the proposed negative feedback of HIF-1a protein on its mRNA abundance?  Please comment on this fascinating finding.

5.     Please add a diagram summarizing the molecular mechanisms of hypoxia-induced APOL1 expression, placing the present findings in the context of current understanding of these mechanisms.

MINOR COMMENTS

1.     Lines 111, 146, 155, 177, 191-192, 238-239: Protein “size” (i.e., mass) markers are not shown, only the numbers.

2.     Lines 107, 143, 149, 179, 196, 287: The correct spelling is “Tukey” as in line 661.

3.     Line 243: “…AB8/13 and IFI16KO podocytes in the presence of roxadustat. The specificity of the cytosolic…”

4.     Line 400: IFI16 is essential for upregulating APOL1 expression in response to hypoxia, but is not required for APOL1-induction by IFNg or STING.

Reviewer 3 Report

Comments and Suggestions for Authors

The manuscript entitled “IFI16 is Indispensable for Promoting HIF-1α-mediated APOL1 Expression in Human Podocytes Under Hypoxic Conditions” addresses the crucial role of IFI16 in mediating HIF-1α-mediated APOL1 Expression in human podocytes under hypoxia. Initially, the authors proved that transient accumulation of HIF-1α in hypoxia upregulates APOL1 expression in podocytes through a cGAS-STING-IRF3-independent pathway. Importantly, IFI16 was shown to selectively affect APOL1 expression, whereas it did not affect other hypoxia-responsive genes in podocytes. In summary, these findings demonstrate the unique contribution of IFI16 to the hypoxia-driven expression of the APOL1 gene under hypoxic conditions. In addition, the data suggest that APOL1 gene expression in hypoxic conditions may be regulated by alternative IFI16-dependent mechanisms. The study appears to be an original one and the findings are interesting.

Comments:     

1) In qRT-PCR (section 4.6.), did the authors check the RNA quality with A260/280 and perform an RT negative control to ensure no DNA contamination in the RNA extraction? Please, add these data in the relevant section in material and methods.

2) The author should mention the amount of RNA used to synthesize cDNA and the amount of cDNA used for qRT-PCR. Please, add these data in the relevant section in material and methods.

3) The author should mention the qRT-PCR condition such as annealing temp. and the number of cycles. Please, add these data in the relevant section in material and methods.

4) The authors are advised to add Table S1 (Primers used in this study) since the provided link for that table within the manuscript in line 663 states “File not found”.

5) In the statistical analysis section, did the authors check data normality before proceeding to one-way ANOVA? Authors are advised to address this point and add the answers in the material and methods section.

6) In Figures 1-5, the authors are advised to describe the number of replicates used in Western blotting in the figure legends. Moreover, was the data extracted from independent samples?

7) In line 107, please double-check the name of the post-hoc test Tukey’s test. It is written as “Tuckey test”.

8) The authors are advised to carefully revise the reference section. The authors are advised to unify the way they write the journal name. Sometimes it is written as an abbreviation (most references) while in reference # 3 it was written as a full name. Please, follow the journal instructions in this regard. 

Comments on the Quality of English Language

Minor editing of the English language is required.
